# Peptidomics as a Tool to Assess the Cleavage of Wine Haze Proteins by Peptidases from *Drosophila suzukii* Larvae

**DOI:** 10.3390/biom13030451

**Published:** 2023-02-28

**Authors:** Wendell Albuquerque, Parviz Ghezellou, Kwang-Zin Lee, Quintus Schneider, Phillip Gross, Tobias Kessel, Bodunrin Omokungbe, Bernhard Spengler, Andreas Vilcinskas, Holger Zorn, Martin Gand

**Affiliations:** 1Institute of Food Chemistry and Food Biotechnology, Justus Liebig University Giessen, Heinrich-Buff-Ring 17, 35392 Giessen, Germany; 2Institute of Inorganic and Analytical Chemistry, Justus Liebig University Giessen, Heinrich-Buff-Ring 17, 35392 Giessen, Germany; 3Fraunhofer Institute for Molecular Biology and Applied Ecology, Ohlebergsweg 12, 35392 Giessen, Germany; 4Institute for Insect Biotechnology, Justus Liebig University Giessen, Heinrich-Buff-Ring 26-32, 35392 Giessen, Germany

**Keywords:** peptidases, peptidomics, *Drosophila suzukii*, thaumatin-like proteins, chitinases, recombinant proteins, wine haze

## Abstract

Thermolabile grape berry proteins such as thaumatin-like proteins (TLPs) and chitinases (CHIs) promote haze formation in bottled wines if not properly fined. As a natural grapevine pest, the spotted-wing fly *Drosophila suzukii* is a promising source of peptidases that break down grape berry proteins because the larvae develop and feed inside mature berries. Therefore, we produced recombinant TLP and CHI as model thermolabile wine haze proteins and applied a peptidomics strategy to investigate whether *D. suzukii* larval peptidases were able to digest them under acidic conditions (pH 3.5), which are typically found in winemaking practices. The activity of the novel peptidases was confirmed by mass spectrometry, and cleavage sites within the wine haze proteins were visualized in 3D protein models. The combination of recombinant haze proteins and peptidomics provides a valuable screening tool to identify optimal peptidases suitable for clarification processes in the winemaking industry.

## 1. Introduction

The formation of turbidity in wines results from the denaturation of thermolabile proteins, which then interact and form cross-links, resulting in particle agglomeration and flocculation at moderate temperatures [1]. Protein aggregation in white wines is also promoted by ingredients such as polyphenols and sulfite ions [2], which link the hydrophobic protein cores and/or rearrange disulfide bridges. The most prominent thermolabile proteins in wine are thaumatin-like proteins (TLPs) and chitinases (CHIs), which are constitutively expressed at a basal level but are induced in response to fungal infections [3]. Accordingly, they are also described as pathogen-related (PR) proteins. Such proteins are stable under acidic conditions and have a compact structure that hinders degradation and facilitates their survival during vinification [4].

Haze formation in bottled wines is prevented by clarification or fining steps during winemaking [5]. Clarification typically involves the addition of bentonite clay, a cation exchanger that binds the wine haze proteins but also negatively affects the organoleptic properties [6,7]. Other fining methods such as the use of animal proteins (casein, gelatin and albumin) and ultrafiltration can be used to remove polyphenolic compounds, but they also reduce the levels of tannin [8].

A promising alternative is the in situ degradation of thermolabile proteins using peptidases. However, peptidase activity is highly dependent on the pH, temperature, and the exposure of buried protein structural features [2,9]. Recently, cleavage sites have been traced along the TLP and CHI polypeptide backbones by high-resolution LC-MS/MS, an approach known as top–down peptidomics [10]. This is based on the analysis of native proteins by high-resolution mass spectrometry (HR-MS) without previous chemical treatments [11], thus preserving their post-translational modifications [12]. Alternatively, when the cleavage profiles of peptidases are unknown, de novo sequencing algorithms [13] can be used to identify peptides and proteins based on peptide fragmentation patterns [14,15].

Although peptidases are ubiquitous and responsible for essential cellular functions in all living organisms, not all peptidases have been studied in detail, particularly those from insects [16,17]. Given their ecological diversity, insects provide an immense and mostly untapped source of novel and uncharacterized proteolytic enzymes [18]. Insects secrete digestive enzymes such as serine, cysteine, metallo-, and aspartate peptidases into their gut or saliva [19,20,21]. The expression of such enzymes in polyphagous insects is influenced by their diverse food sources [22,23]. For example, the spotted-wing fly *Drosophila suzukii* (Diptera: Drosophilidae) is an invasive pest of fruit crops that has adapted to thrive under many environmental conditions [24]. It causes significant crop losses by laying eggs in ripe or ripening fruits such as cherries, plums, and grape berries [25]. Indeed, *D. suzukii* infests ripe grape berries to exploit the increasing sugar content, lower pH, and lower skin penetration resistance [26]. Because *D. suzukii* larvae can survive in this acidic environment, they offer a promising source of acidic peptidases that remain active under typical winemaking pH conditions.

The application of peptidases with residual activity under acidic conditions could facilitate wine clarification. Therefore, we extracted peptidases from *D. suzukii* larvae and characterized the purified enzymes to evaluate their ability to cleave haze-forming proteins at pH 3.5. Recombinant TLP and CHI (rTLP and rCHI) were expressed, purified [27], and digested with the larval peptidases at pH 3.5. The cleaved peptides were detected by LC-MS/MS analysis, and the resulting peptidomics data were analyzed by de novo sequencing. The cleavage sites were visualized using 3D protein models. Finally, we tested the potential of the purified peptidases to degrade native wine proteins isolated from a Silvaner Franken wine in vitro. The combination of methods described herein would be suitable for screening enzyme candidates that can be used as haze preventers in the wine industry.

## 2. Materials and Methods

### 2.1. Wine Proteins

We expressed rTLP and rCHI, each bearing a His_6_-tag, in the yeast *Komagataella phaffii* (formerly *Pichia pastoris*). Plasmid vectors (described in Albuquerque et al. [27]) containing the genes encoding a thermolabile TLP (4JRU isoform) and a class IV CHI were introduced into *K. phaffii* cells by electroporation. The yeast cells were cultured for 4 days in buffered methanol-complex medium (BMMY) with a daily feed of 1% methanol. The proteins were recovered from the culture medium and purified by chromatography, as previously described [27]. Briefly, the culture supernatant was loaded onto a 5-mL HiTrap IMAC FF column (Cytiva, Freiburg, Germany) and eluted with 0.2 M phosphate buffer (pH 6.0) containing 250 mM imidazole (Carl Roth, Karlsruhe, Germany) directly into a HiLoad 16/600 Superdex 75 column (Cytiva) mounted on an NGC fast protein liquid chromatography (FPLC) system (Bio-Rad Laboratories, Munich, Germany). Size exclusion chromatography (SEC) was carried out at a flow rate of 1 mL/min in 0.1 M Tris-HCl (pH 7.0). The eluted proteins were quantified as described by Bradford [28]. The proteins were separated by SDS-PAGE (sodium dodecyl sulfate–polyacrylamide gel electrophoresis) in a 12% polyacrylamide gel under denaturing conditions [29] and visualized by staining with Coomassie Brilliant Blue (AppliChem, Darmstadt, Germany) and by Western blot (Bio-Rad Laboratories) using antibodies specific for His_6_-tag (Thermo Fisher Scientific GmbH, Bremen, Germany). The tag was then removed by incubating the recombinant proteins with tobacco etch virus peptidase (Biozol Diagnostica, Eching, Germany) for 2 h at 37 °C. The proteins were stored at 4 °C. Colloids were obtained from a Silvaner Franken wine by ultrafiltration, as described by Albuquerque et al. [10].

### 2.2. Rearing of Drosophila suzukii

Adult *D. suzukii* flies were reared on a sterile soymeal and cornmeal medium comprising 9% (*w*/*v*) soymeal and cornmeal mix, 1.8% (*w*/*v*) brewer’s yeast, 0.8% (*w*/*v*) agar, 8% (*w*/*v*) malt, 2.2% (*w*/*v*) molasses, 0.2% (*w*/*v*) nipagin in 70% ethanol. and 0.625% (*v*/*v*) propionic acid in distilled water and kept in a climate chamber (Regineering, Preith, Germany) at 26 °C, 60% relative humidity, with a 12-h photoperiod. *D. suzukii* eggs were kept in the growth medium before hatching, and the larvae (Figure 1) were harvested after 9–11 days by floating on a solution of 50 mM sucrose. Larval samples were rinsed in tap water on a 300 mic test sieve (Retsch, Haan, Germany) and stored at –80 °C in reaction tubes for further experiments.

### 2.3. Extraction of Peptidases from D. suzukii Larvae

Frozen larval samples (estimated at 4000 larvae per extraction) were macerated under liquid nitrogen and the homogenate was incubated overnight at 4 °C in 20 mL of extraction buffer (0.1 M Tris-HCl pH 7.0, 0.15 M NaCl, and 1% Triton X-100) for cell lysis. The cell debris was removed by centrifugation at 12,000× *g* (for 10 min at 4 °C) using an Allegra X-15R device (Beckman Coulter, Brea, CA, USA) and the supernatants were concentrated in microcentrifuge tubes with a molecular weight cut-off (MWCO) of 10 kDa (Merck, Darmstadt, Germany).

### 2.4. Proteolytic Activity

The activity of the extracted peptidases was quantified by a rapid spectrophotometric method (spectrophotometer BioTek Synergy, Agilent Technologies, Santa Clara, CA, USA) using azocasein as a substrate and according to Leighton et al. [30]. The degradation of native wine proteins was analyzed by the agar diffusion assay [31]. Briefly, we prepared 1% (*w*/*v*) agarose (Biozym Scientific, Oldendorf, Germany) in 0.1 M citrate buffer (pH 3.5) by boiling and then cooling to 50 °C before mixing with the proteins (1 mg/mL in 0.1 M citrate buffer, pH 3.5) isolated from a Silvaner Franken wine. The solution was poured (20 mL) into Petri dishes and 1 mM ampicillin was added to avoid microbial contamination. Holes measuring 1 cm in diameter (made with a sterile scalpel) were punched into the polymerized agar gels before adding 20 μL of the cell lysate or purified peptidase. The plates were incubated overnight at 37 °C and stained with Coomassie Brilliant Blue to detect halos. Zymograms were prepared by embedding casein (Carl Roth) in SDS-PAGE gels under semi-native conditions (pH 8.8). Degradation bands were observed after staining the gels with Coomassie Brilliant Blue.

### 2.5. Purification and Characterization of Peptidases from D. suzukii Larvae

#### 2.5.1. Chromatography

Peptidases in the cell lysates were purified by anion-exchange (AEX) chromatography on a HiPrep DEAE FF column 16/10 (GE Healthcare, Chicago, IL, USA) and SEC on a Superdex 75 column 10/300 GL (GE Healthcare) mounted on the NGC FPLC system. AEX chromatography was carried out at a flow rate of 1 mL/min and the proteins were eluted by isocratic flow in a mixture of buffer A (0.1 M Tris-HCl pH 7.0) and buffer B (0.1 M Tris-HCl pH 7.0 containing 0%, 20%, 50% or 100% 1 M NaCl). For SEC, the samples were eluted at 1 mL/min using an isocratic flow of 0.1 M Tris-HCl (pH 7.0) containing 0.15 M NaCl. The azocasein and agar diffusion assays described above were used to identify peaks containing peptidase activity. The protein samples were pooled, desalted by dialysis in float-A-Lyzer devices (in 0.1 M Tris-HCl, pH 7.0, under magnetic stirring and with a membrane with MWCO of 10 kDa, LubioScience, Zürich, Switzerland) and concentrated in microcentrifuge tubes with a MWCO of 10 kDa. Conductivity was monitored during each chromatographic step to ensure the complete desalting of the samples. Protein stability was preserved by keeping the samples at 4 °C during chromatography and on ice during sample handling.

#### 2.5.2. LC-MS/MS Analysis

The protein fractions from each chromatography step were digested with trypsin (Promega, Madison, WI, USA) and the peptides were concentrated and desalted using C18 ZipTip pipette tips (Merck). The peptides were then separated on a Kinetex C18 column (2.1 mm × 100 mm, 2.6 µm, 100 Å; Phenomenex, Torrance, CA, USA) mounted on a Dionex UliMate 3000 RSL UHPLC system (Thermo Fisher Scientific GmbH). This was connected to a Q Exactive HF-X orbital trapping mass spectrometer (Thermo Fisher Scientific GmbH). LC-MS/MS was carried out using the parameters described by Albuquerque et al. [10]. The MS raw data were used to screen the UniProt database (taxonomically restricted to *Drosophila suzukii*) using Byonic v4.2 (Protein Dynamics, Cupertino, CA, USA) and Proteome Discoverer v2.2 (Thermo Fisher Scientific GmbH). The following parameters were applied: two missed cleavage sites; minimum peptide length six amino acids; MS1 and MS2 tolerances of 10 ppm and 0.5 Da, respectively. A strict target FDR (false discovery rate) of 0.01 and a relaxed target FDR of 0.05 were used to validate the identified peptide-spectrum matches and to filter the final output.

#### 2.5.3. Gene Expression Analysis by Quantitative RT-PCR

The gene sequences matching the identified peptidases were obtained from a public transcriptome dataset of *D. suzukii* (NCBI database) and were amplified by quantitative real-time reverse transcription polymerase chain reaction (RT-PCR). *D. suzukii* total RNA was isolated using TRI Reagent (Zymo Research, Freiburg, Germany) and reverse transcribed using the iScript cDNA Synthesis Kit (Bio-Rad). The resulting cDNA was amplified using Power SYBR Green PCR Master Mix (Thermo Fisher Scientific GmbH) and the primers listed in Table 1. A gene encoding actin was used as the reference gene.

### 2.6. Analysis of Cleavage Sites in rTLP and rCHI

Purified and vacuum-dried rTLP and rCHI were dissolved in 0.1 M citrate buffer (pH 3.5) to a concentration of 0.25 mg/mL and incubated (in a heating-thermomixer HLC, DITABIS AG, Pforzheim, Germany) with the purified peptidases at 37 °C for 18 h. The cleaved peptides were collected by filtration by using Amicon filters with a MWCO of 10 kDa. Native peptides produced by the digestion of rTLP and rCHI with *D. suzukii* larval peptidases were identified by LC-MS/MS (peptidomics). Raw MS data were analyzed using Peaks Studio vX+ (Bioinformatics Solutions, Waterloo, ON, Canada) based on de novo sequencing [32]. The obtained sequences were searched against the UniProtKB databases (and further correlated to NCBI accession numbers) taxonomically restricted to *Vitis vinifera.* Protein visualization and cleavage site analysis were carried out using Pymol v2.0 (Schrödinger, New York, NY, USA) with 3D models of thermolabile TLP (PDB code 4JRU) and a class IV CHI homology model [10]. For control analysis, rTLP and rCHI were filtered without degradation (incubated at the same pH and temperature but without peptidases) to confirm that no digested peptides were produced without enzymatic reactions.

## 3. Results

### 3.1. Purified Peptidases

Five protein peaks were detected in the AEX chromatogram of the *D. suzukii* larval cell lysates (Figure 2(aI)). The proteolytic activity of peak 3 (eluted with an isocratic flow of 20% 1 M NaCl) showed a clear halo of degradation around the wine proteins embedded in agar at pH 3.5 (Figure 2(aII)) and the specific activity against azocasein was 6674.8 U/mg. After desalting, fraction 3 from the AEX step was pooled for SEC, resulting in six further peaks (Figure 2(bI)). Fraction C showed a visible halo of degradation around the wine proteins in the agar diffusion assay at pH 3.5 (Figure 2(bII)) and the specific activity toward azocasein was 8845 U/mg. Moreover, we observed distinct degradation bands in a casein zymogram (Figure 2c) from ~38 kDa to ~180 kDa. The protein composition after each chromatography step was compared by electrophoresis (Figure 2d), and the degree of purification was calculated in terms of the yield and purification factor (Figure 2e). The proteins identified by MS are shown in Figure 2(aIII,bIII).

### 3.2. Characterization of Peptidases by MS-Based Proteomics

Twelve unique peptidases were identified in the purified protein fractions (in both AEX and SEC), each representing different levels of protein coverage (Table 2). We identified candidates representing the glutamyl aminopeptidase-like (NCBI: XP_016943864.1), caspase-3 (XP_016923550.1), Xaa-Pro dipeptidase (XP_016941450.2), serine protease 1/2-like (XP_016934104.1), and venom serine protease (XP_016935480.1) families after both chromatography steps, which provide additional confidence in the results. We identified aminopeptidase N (XP_016935991.1), chymotrypsin 1 (XP_016924069.1), and trypsin-7 (XP_016930621.1) exclusively after the AEX step, whereas γ-glutamyltranspeptidase 1 (XP_016930772.1), dipeptidyl peptidase 3 (XP_016925042.1), serine protease 42-like (XP_016940780.1), and cathepsin L1 (XP_016943011.1) proteins were identified exclusively after the SEC step. The identification of three unique peptides representing glutamyl aminopeptidase-like (XP_016943864.1, coverage 4%) and serine protease 1/2-like proteins (XP_016934104.1, coverage 5%) as well as γ-glutamyltranspeptidase 1 (XP_016930772.1, coverage 4.1%) increased our confidence in their identification following the SEC step. A list of all identified peptidases and their closest relatives is provided in Appendix A.

### 3.3. Gene Expression Analysis

The expression profiles of genes encoding the identified glutamyl aminopeptidase-like (metallopeptidase), caspase-3 (cysteine peptidase), cathepsin L1, dipeptidyl peptidase III (metallopeptidase), and a serine protease 1-like (serine peptidase) proteins are summarized in Appendix A.

### 3.4. Identification of Intact Peptides (LC-MS/MS Top–Down Proteomics)

The rTLP and rCHI proteins were enzymatically hydrolyzed at pH 3.5 under native conditions (without the use of denaturing agents) by peptidases from the *D. suzukii* larvae. The peptides detected by MS are highlighted in the 3D structures of rTLP and rCHI, which show the surface features and secondary structures alongside the anticipated trypsin cleavage pattern at pH 7.0 (Figure 3). Only one peptide cleavage product was found for rTLP, whereas the structure of rCHI was largely degraded. The secondary structures cleaved by the purified peptidases are shown in Figure 3(aI,bI) for rTLP and rCHI, respectively. The cleavage sites are also highlighted in the corresponding amino acid sequences.

## 4. Discussion

We identified at least 12 peptidases from the *D. suzukii* larvae with molecular masses ranging from 28.2 to 108.5 kDa. This number of peptidases may explain the complex pattern of degradation bands observed in the casein zymograms (Figure 2c). The purified peptidases were assigned to three different classes based on their active sites: five serine peptidases (trypsin-7, chymotrypsin 1, venom serine protease, serine protease 1/2-like, and serine protease 42-like), four metallopeptidases (glutamyl aminopeptidase-like, aminopeptidase N, Xaa-Pro dipeptidase, dipeptidyl peptidase 3), and three cysteine peptidases (caspase-3, γ-glutamyltranspeptidase 1, and cathepsin L1).

The *D. suzukii* peptidases were able to cleave rTLP and rCHI under acidic conditions (pH 3.5) at 37 °C, and the resulting peptides were detected by MS-based peptidomics. The use of purified recombinant proteins rather than a complex mixture of wine proteins made it possible to study the direct action of the peptidases on the structure of haze proteins. TLP is a compact protein because it features several disulfide bridges and β-sheet secondary structures [3]. In contrast, class IV CHI is mainly composed of α-helices and loops [33], which facilitates its irreversible denaturation at temperatures above 50 °C [34]. As a consequence, rCHI was completely degraded by the purified *D. suzukii* peptidases, producing ~45 distinct peptides (Figure 3b, Appendix A), suggesting that the rCHI structure was more accessible to the peptidases under native conditions (37 °C). In contrast, the rTLP structure was only cleaved at one specific site, releasing the peptide NVNAGTTGGRVW (Figure 3a, Appendix A).

Not every *D. suzukii* larval peptidase is likely to be active at pH 3.5, and the cleavage of rTLP and rCHI therefore probably reflects the combined action of the acidic peptidases. Furthermore, the metabolic function of each peptidase must be taken into account when considering their potential to cleave wine haze proteins efficiently. Digestive peptidases, for example, are typically endopeptidases with broad cleavage specificity [35]. Some of the identified peptidases have a luminal digestive function including trypsin-7 [36], chymotrypsin 1 [37], Xaa-Pro dipeptidase or prolidase [38], and cathepsin L1 [39].

Serine peptidases are essential digestive enzymes in the insect gut [40,41] and are supported by cysteine peptidases and others as an evolutionary strategy to overcome the production of serine peptidase inhibitors by plants [42]. Serine proteases are typically endopeptidases with optimal activity under neutral to alkaline conditions (pH 7–10) at moderate temperatures (20–50 °C), and they have distinct functions in insect development, reproduction, and metabolism [43,44,45]. Many other insect metallopeptidases are classified as aminopeptidases, which act as *N*-terminal exopeptidases [46,47]. These enzymes favor alkaline pH conditions and temperatures of 30–60 °C [48,49,50]. In contrast, cysteine peptidases have an optimal pH range of 4–6 [51,52] and they complement serine peptidases in insect nutrition [42]. Among the *D. suzukii* larval peptidases we identified, γ-glutamyltranspeptidase 1 (XP_016943864.1), caspase-3 (XP_016923550.1), and cathepsin L1 (XP_016943011.1) are cysteine peptidases, which in *Drosophila* species have been associated with apoptosis and the digestion of cytoplasmic components. Caspase-3 is a cysteine aspartic peptidase (cleaving after aspartic acid residues) that participates in death signaling and apoptosis [53]. Caspases are most active at pH~4, implying that they are located in vacuoles rather than the cytosol [54]. In *Drosophila* cells, the degradation of filamentous actin, α-tubulin, α-spectrin, and nuclear lamins coincides with caspase-3 activity [55]. Furthermore, γ-glutamyltranspeptidase 1 is a transmembrane glycoprotein [56] catalyzing the transpeptidation and hydrolysis of the γ-glutamyl group of glutathione [57]. This is achieved by cleaving the γ-glutamyl bond (γ-Glu-Cys-Gly), releasing free glutamate and the dipeptide cysteinyl-glycine [58]. This enzyme also regulates apoptosis depending on the levels of intracellular glutathione [59]. The expression and activity of bacterial γ-glutamyltranspeptidase was found to be induced at pH ~4 by the addition of glutamine and salts [56].

Cathepsins belong to the papain family and feature a cysteine residue in their active site [60]. They are active at pH 3–4.5 [61] and have a broad specificity for protein cleavage sites according to the MEROPS database [62]. This broad activity was confirmed by using nano-LC-MS/MS to measure the frequency and distribution of cleavage sites when *Fasciola hepatica* cathepsin L1 was used for the complete degradation of hemoglobin [61]. Cathepsin L1 is a digestive peptidase in many organisms, transforming proteins into absorbable peptides [63]. Insects secrete cathepsins from epithelial cells into the gut [64], although they are only active in acidic regions [65]. For example, cathepsin L1 has been described as an acidic endopeptidase that is unstable at neutral pH [66]. The cathepsin L1 we identified in the *D. suzukii* larvae may be responsible for the observed cleavage profile of rTLP and rCHI (Figure 3), given its broad specificity as a typical digestive enzyme.

The cleavage of rTLP and rCHI should also be tested under winemaking conditions, specifically a low pH and temperature, high ionic strength, and adequate concentrations of ethanol, sulfite, and polyphenols. Therefore, the following workflow should be implemented: (1) expression of recombinant enzyme candidates and scaled-up production for the most promising peptidases such as cathepsin L1; (2) purification of the peptidases using appropriate tags; and (3) evaluation of their ability to reduce haze formation in real wine samples. A peptidase (or a mixture) that reduces wine haze under typical winemaking conditions would constitute a real breakthrough in wine research. Innovative methods for haze prevention are still in demand. Recently, the use of an enzymatic mixture known as Proctase combined with flash pasteurization was approved by the International Organization of Vine and Wine (OIV) for market applications [67]. Furthermore, the combination of rTLP and rCHI and the identification of cleavage sites by MS-based peptidomics and de novo sequencing is an innovative tool to screen peptidase candidates for their ability to cleave thermolabile grape berry proteins at low pH and temperatures suitable for winemaking, providing more insights into the effects of peptidases in wine haze reduction. Our workflow could remove hurdles preventing the use of peptidases in industrial wine fining.

## 5. Conclusions

Peptidases purified from the *D. suzukii* larvae were identified by MS-based proteomics and evaluated regarding their potential to cleave wine proteins under acidic conditions. The identified peptidases cleaved recombinant rTLP and rCHI proteins at pH 3.5, and the digestion products were detected by top–down MS-based peptidomics. Acid peptidases such as cathepsin L1 are likely to be responsible for the observed cleavage profile given their activity at low pH, broad cleavage specificity, and natural function as digestive enzymes. The methods discussed herein can be used to screen for peptidases that are optimal for eventual winemaking applications.

## Figures and Tables

**Figure 1 biomolecules-13-00451-f001:**
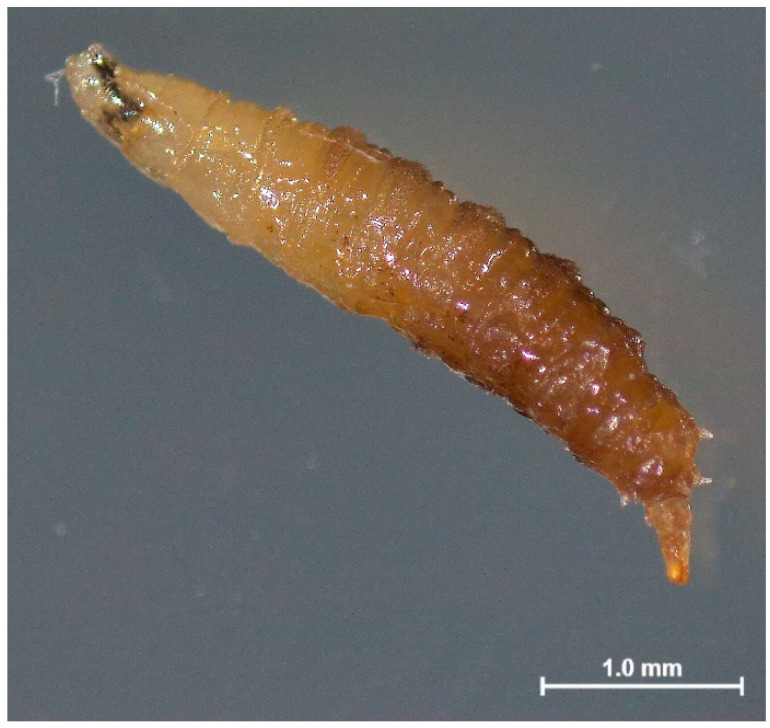
Microscopic image of *Drosophila suzukii* (larval stage).

**Figure 2 biomolecules-13-00451-f002:**
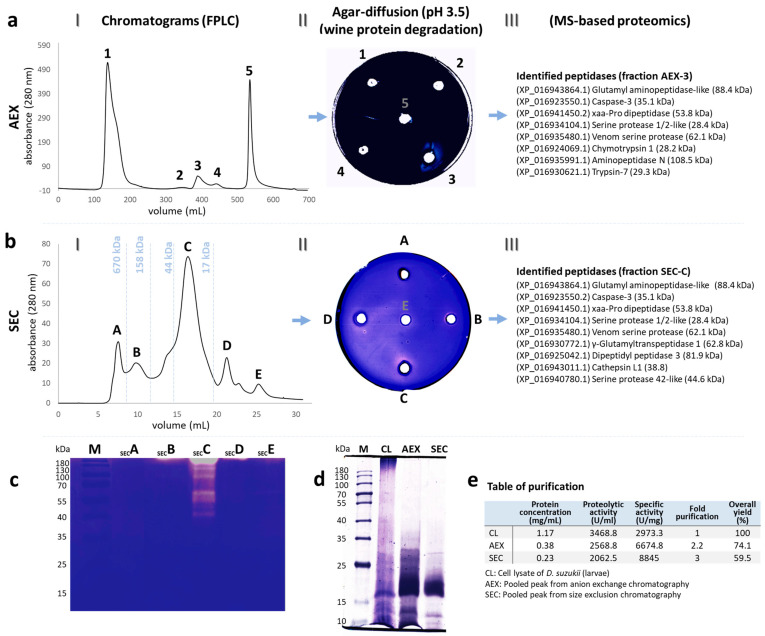
Partial purification of peptidases from the *D. suzukii* larvae. (**aI**) Anion exchange chromatography (AEX) of the cell lysates (CL) of the *D. suzukii* larvae. (**aII**) The proteolytic activity of each peak was assessed by the degradation of proteins from a Silvaner Franken wine in an agar diffusion assay. (**aIII**) Peptidases in peak III identified by MS-based proteomics. (**bI**) Size exclusion chromatography (SEC) of peak 3 from the AEX step. (**bII**) Degradation of wine proteins assessed by the agar diffusion assay. (**bIII**) Peptidases in peak C identified by MS-based proteomics. (**c**) Protein peaks purified by SEC tested by casein zymography. (**d**) SDS-PAGE analysis to characterize protein separation during each purification step. (**e**) Table of purification efficiency and comparison of the activity and protein content during the purification process.

**Figure 3 biomolecules-13-00451-f003:**
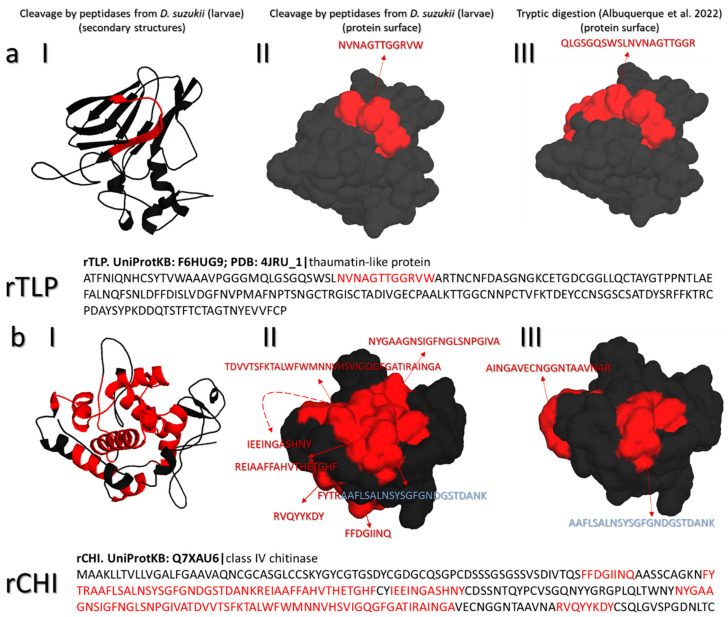
Cleavage sites in rTLP and rCHI detected after digestion with the purified peptidases from the *D. suzukii* larvae. The cleaved peptides from (**a**) rTLP and (**b**) rCHI are displayed separately as (**I**) cleaved peptides identified in secondary structures and (**II**) cleaved peptides identified on the protein surface. (**III**) The same protein structure is shown cleaved by a trypsin. For each recombinant protein, the identified peptides are also highlighted in red in the amino acid sequence.

**Table 1 biomolecules-13-00451-t001:** Primers used to analyze the expression of genes encoding the *D. suzukii* larval peptidases.

Peptidase	ID (NCBI)	Forward Primer	Reverse Primer
Glutamyl aminopeptidase-like	XP_016943864.1	5′-TGTGCATCATTGTGTCCGAC-3′	5′-TCGATCTGATGGGAAGTGGC-3′
Caspase-3	XP_016923550.1	5′-GACTGCCAGGACGCCAAC-3′	5′-CGCTCGCAATTCTCGTATGT-3′
Serine protease 1/2-like	XP_016934104.1	5′-GCGACAACACTATCTGCACC-3′	5′-CTGACTCCCACCAGCTTGTT-3′
Dipeptidyl peptidase III	XP_016925042.1	5′-CGAGCACTACATCCGATCCT-3′	5′-TCCCTTGTCCTTGATCCACC-3′
Cathepsin L1	XP_016943011.1	5′-CAACTGCAATCGTTCCCCAA-3′	5′-TCGTCCGAGTATACCTTGCC-3′

**Table 2 biomolecules-13-00451-t002:** Peptidases from the *D. suzukii* larvae identified following the AEX and SEC purification steps (entries in bold were identified after both steps). The identification score is represented in terms of the percent coverage and unique peptides. The identified enzymes and their peptides detected by LC-MS/MS are related to the NCBI accession number. The catalysis type and molecular masses are provided for each enzyme based on the NCBI data.

n	Accession No. (NCBI)	Description	Peptides	Type	Coverage [%]	Unique Peptides	MW (kDa)
**AEX step**							
1	XP_016943864.1	**Glutamyl aminopeptidase-like**	R.QAFPCFDEPALK.A K.YNIEWLAR.N K.WWNDLWLNEGFAR.F	Metallo	4	3	88.4
2	XP_016923550.1	**Caspase-3**	R.TYDDLTFSDINDK.L	Cysteine	4	1	35.1
3	XP_016941450.2	**Xaa-Pro dipeptidase**	K.SLYNTDVDYVFR.Q	Metallo	2	1	53.8
4	XP_016934104.1	**Serine protease 1/2-like**	K.VELPSYNDR.Y	Serine	5	1	28.4
5	XP_016935480.1	**Venom serine protease**	K.FLQQDFVGMNPFVAGWGAVK.H	Serine	4.1	1	62.1
6	XP_016924069.1	Chymotrypsin 1	R.ILGGEDVEQGEYPWSASVR.Y	Serine	8.6	1	28.2
7	XP_016935991.1	Aminopeptidase N	K.QLIDPIFNK.I	Metallo	1	1	108.5
8	XP_016930621.1	Trypsin 7	R.EWLEETIEANK.D	Serine	4	1	29.3
**SEC step**							
1	XP_016943864.1	**Glutamyl aminopeptidase-like**	R.QAFPCFDEPALK.A K.YNIEWLAR.N K.WWNDLWLNEGFAR.F	Metallo	4	3	88.4
2	XP_016923550.1	**Caspase-3**	R.TYDDLTFSDINDK.L	Cysteine	4	1	35.1
3	XP_016941450.2	**Xaa-Pro dipeptidase**	K.SLYNTDVDYVFR.Q	Metallo	2	1	53.8
4	XP_016934104.1	**Serine protease 1/2-like**	K.VELPSYNDR.Y	Serine	5	1	28.4
5	XP_016935480.1	**Venom serine protease**	K.FLQQDFVGMNPFVAGWGAVK.H	Serine	4.1	1	62.1
6	XP_016930772.1	γ-Glutamyltranspeptidase 1	R.YGILPWK.R R.LFEPSIK.L K.EIYDGGETGR.K	Cysteine	4.1	3	62.8
7	XP_016925042.1	Dipeptidyl peptidase 3	K.IFDK.V	Metallo	2	2	81.9
8	XP_016943011.1	Cathepsin L1	R.LGVNPLADMTR.K	Cysteine	3.1	1	38.8
9	XP_016940780.1	Serine protease 42-like	K.DGEYQVILK.K K.LWNIDPK.Y	Serine	3.9	2	44.6

## Data Availability

The original contributions presented in the study are included in the article and Appendix A. Further inquiries can be directed to the corresponding author.

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
