# Peer review of "Peptidomics as a Tool to Assess the Cleavage of Wine Haze Proteins by Peptidases from Drosophila suzukii Larvae"

_biomolecules, 2023, doi:10.3390/biom13030451_

Round 1
Reviewer 1 Report
The researchers have used purified recombinant proteins to study the direct action of the peptidases on the structure of haze forming proteins found in some wines. Confirming this activity in actual wine making process would be an interesting future study.
Section 2.4. What is the reasoning behind using spectrophotometry and not other more specific and sensitive techniques (e.g fluorescent or radioactive peptide substrates)? How did you monitor loading volume of the agar? How were the holes punched into the agar? Define the acronym SDS-PAGE.
Section 2.5.1. Please elaborate on how you maintained the purity and stability of the desired proteins during purification. What steps were taken to reduce contamination? What specific yields did you achieve of the target molecules?
Section 2.5.2 How did you confirm that the samples were effectively desalted? Did you use a reference for this? Please elaborate.
Section 2.5.3. What reference gene was used for the quantitative RT-PCR?
Section 2.6 Describe how they were incubated and the equipment that you used.
Discussion:
Cathepsin L1 is a lysosomal cysteine protease and in acid conditions its activity is reduced. Please elaborate on the optimal conditions for cathepsin L1 and how pH reduces its activity.
Conclusion:
I suggest rephrasing the sentence “The methods discussed herein…” This study has not been completed in winemaking style conditions and the results only relate to a specific use case.
Author Response
The researchers have used purified recombinant proteins to study the direct action of the peptidases on the structure of haze forming proteins found in some wines. Confirming this activity in actual wine making process would be an interesting future study.
Section 2.4. What is the reasoning behind using spectrophotometry and not other more specific and sensitive techniques (e.g fluorescent or radioactive peptide substrates)?
We used a rapid and quantitative spectrophotometric method to identify overall peptidase activity based on casein degradation (using azocasein as substrate). Initially, the peptidases obtained from insect cell lysates were not known, therefore the use of a rapid quantitative method proved to be effective and helped to identify the protein fractions that contained peptidases. We modified the text and added the information that it was a quantitative and fast method based on Leighton et al. (1973) in the lines 135f.
How did you monitor loading volume of the agar?
A volume of 20 mL of an agar-containing solution was added to each Petri dish. This information was added to the manuscript in line 142.
How were the holes punched into the agar?
The holes were made with a sterile scalpel. This information was added to the manuscript in line 143.
Define the acronym SDS-PAGE.
The information was provided in section 2.1 in line 106.
Section 2.5.1. Please elaborate on how you maintained the purity and stability of the desired proteins during purification. What steps were taken to reduce contamination?
To reduce contamination, the cell lysates were centrifuged and filtered as described in section 2.3. For the purification process, the columns were previously washed with 1 M NaCl to reduce contaminants from successive chromatographic runs. The stability of the samples was preserved by maintaining the samples at 4 °C during the chromatographic steps and on ice during sample handling, information which was added in lines 164f.
What specific yields did you achieve of the target molecules?
The overall yield and the purification degree are presented in figure 2f.
Section 2.5.2 How did you confirm that the samples were effectively desalted? Did you use a reference for this? Please elaborate.
Conductivity was monitored during the purification process in the FPLC systems to ensure the complete desalting of the samples. This information was added to section 2.5.2 in lines162f.
Section 2.5.3. What reference gene was used for the quantitative RT-PCR?
A gene encoding actin was used as reference. The information was added to the manuscript in line 188.
Section 2.6 Describe how they were incubated and the equipment that you used.
The samples were incubated in a thermomixer (HLC, DITABIS AG, Pforzheim, Germany). The information was added to the manuscript in the lines 193-194.
Discussion:
Cathepsin L1 is a lysosomal cysteine protease and in acid conditions its activity is reduced. Please elaborate on the optimal conditions for cathepsin L1 and how pH reduces its activity.
The following information was provided in the discussion section in lines 332-333 “Cathepsin L1 is an acidic endopeptidase that is unstable at neutral pH [66].” Based on this literature from Turk et al. 2012, increased pH (basic pH values) would reduce the activity of the enzyme Cathepsin L1.
Conclusion:
I suggest rephrasing the sentence “The methods discussed herein…” This study has not been completed in winemaking style conditions and the results only relate to a specific use
We fully agree with this comment, and we have modified the following sentence in lines 360-361:
“The methods discussed herein can be used to screen for peptidases that are optimal for eventual winemaking applications”
Reviewer 2 Report
Dear authors,
I find you research very interesting and well-written. The subject of this manuscript deals with the issue of wine haze and gives a new solution for it, using peptidases from D. suzukii larve. The introduction provides enough information, the methods are adequately described and results are clearly presented and supported by discussion.
I recommend to change the reference style according to the instructions for this journal (numerals in square brackets) and add them in reference list according to their appearance in the text.
Other than that, it is my opinion that this manuscript is well-written and scientifically sound, and it could be published in this special issue.
Author Response
Dear authors,
I find you research very interesting and well-written. The subject of this manuscript deals with the issue of wine haze and gives a new solution for it, using peptidases from D. suzukii larvae. The introduction provides enough information, the methods are adequately described and results are clearly presented and supported by discussion.
I recommend to change the reference style according to the instructions for this journal (numerals in square brackets) and add them in reference list according to their appearance in the text.
Other than that, it is my opinion that this manuscript is well-written and scientifically sound, and it could be published in this special issue.
Thank you very much for your positive feedback. As requested, we have changed the reference style according to the journal’s instructions as well as the reference list.
Reviewer 3 Report
Dear Authors
The article is well written and contains an interesting experiment. I consider the results and discussions useful for the winemaking process.
I have no other comments!
Author Response
Dear Authors
The article is well written and contains an interesting experiment. I consider the results and discussions useful for the winemaking process.
I have no other comments!
Thank you very much for your positive evaluation of our work!
Reviewer 4 Report
Dear Authors,
I have some suggestions to improve their work.
In the text, the citation of the articles does not correspond to what was requested by the journal.
The references are not in the format and order suggested.
Please, check the author's guidelines.
A grammatical revision is required throughout the text; for example, some articles, commas, adverbs, Etc., must be included.
The manuscript needs to screen for typos and grammatical errors. Keep the same font style and size throughout the manuscript.
Some significant points should be addressed before the manuscript can be considered for publication.
Methods
The plasmids name of recombinant production is missing.
The brand of some reagents and equipment used need to be included, for example, the anti-His antibody.
In the Extraction of peptidases from the D. suzukii larvae section, some details are missing; for example, the quantity (mg or g)or numbers of D. suzikii larvae, the centrifugation time and temperature are not indicated in the protocol of endogenous protein recuperated.
In the Purification and characterization of peptidases from D. suzukii larvae, the Molecular weight cut-off specifications and rates of buffer exchange for the dialysis are missing.
The structure analysis and modeling need to be included.
Authors need to mention which software or tool was used for sequence analysis and comparison in identified peptides.
Results:
The author says: that the expression profiles of genes encoding the identified glutamyl aminopeptidase-like (metallopeptidase), caspase-3 (cysteine peptidase), cathepsin L1, dipeptidyl peptidase III (metallopeptidase) and a serine protease 1-like (serine peptidase) proteins are summarized in Supplementary Data, but is missing the information.
Is the gene expression of proteins identified correlated with the protein expression and activity?
Change the number of tables.
Author Response
Dear Authors,
I have some suggestions to improve their work.
In the text, the citation of the articles does not correspond to what was requested by the journal.
The references are not in the format and order suggested.
Please, check the author's guidelines.
The citations throughout the manuscript and the references were edited as suggested.
A grammatical revision is required throughout the text; for example, some articles, commas, adverbs, Etc., must be included.
The complete article was carefully checked and minor errors were corrected.
The manuscript needs to screen for typos and grammatical errors. Keep the same font style and size throughout the manuscript.
Fonts and sizes were checked and unified accourding to the journal’s instructions.
Some significant points should be addressed before the manuscript can be considered for publication.
Methods
The plasmids name of recombinant production is missing.
The information about the plasmid is available at Albuquerque et al. 2022. Recombinant thaumatin-like protein (rTLP) and chitinase (rCHI) from Vitis vinifera as Models for Wine Haze Formation. Molecules. Molecules 2022, 27(19), 6409; https://doi.org/10.3390/molecules27196409.
This information was added to the manuscript in line 94.
The brand of some reagents and equipment used need to be included, for example, the anti-His antibody.
The information about the manufacturer of the anti-His antibody (Thermo Fisher Scientific GmbH, Bremen, Germany) and the spectrophotometer (BioTek Synergy, Agilent Technologies, Santa Clara, CA, USA) was added in lines 109-110 and 136-137.
In the Extraction of peptidases from the D. suzukii larvae section, some details are missing; for example, the quantity (mg or g) or numbers of D. suzukii larvae.
We added the information, that roughly 4,000 larvae (per 20 mL of extraction buffer) were used. This information was added to the manuscript in line 127.
The centrifugation time and temperature are not indicated in the protocol of endogenous protein recuperated.
The information on the centrifugation time (10 min) and the temperature (4 °C) was added in lines 130.
In the Purification and characterization of peptidases from D. suzukii larvae, the Molecular weight cut-off specifications and rates of buffer exchange for the dialysis are missing.
Dialysis was performed in float-A-Lyzer devices (in 0.1 M Tris-HCl, pH 7.0, under magnetic stirring and with a membrane with MWCO of 10 kDa) The information about the molecular mass cut-off (MWCO) of 10 kDa regarding to the dialysis device was added in lines 160f.
The structure analysis and modeling need to be included. Authors need to mention, which software or tool was used for sequence analysis and comparison in identified peptides.
Sequences analysis was obtained from the softwares Peaks Studio and the structural correlation was made by visualization in the software Pymol v2.0. Both of the softwares were described in the section 2.6.
Results:
The author says: that the expression profiles of genes encoding the identified glutamyl aminopeptidase-like (metallopeptidase), caspase-3 (cysteine peptidase), cathepsin L1, dipeptidyl peptidase III (metallopeptidase) and a serine protease 1-like (serine peptidase) proteins are summarized in Supplementary Data, but is missing the information.
We apologize, that we did not upload the Supplementary Data 1 and 2 in the original submission; in the revision both files have been uploaded.
Is the gene expression of proteins identified correlated with the protein expression and activity?
The amplification of the genes encoding peptidases was performed to ensure the existence of the genetic information regarding the identified peptidases in the genome of D. suzukii. Enzyme activity could not be correlated to expression levels since the peptidases were not individually purified or recombinantly expressed.
Change the number of tables.
The numbers were corrected.